# Visual working memories are abstractions of percepts

**Ziyi Duan[1], Clayton E Curtis[1,2]***

[1]Department of Psychology, New York University, New York, United States; [2]Center for Neural Science, New York University, New York, United States

**Abstract** During perception, decoding the orientation of gratings depends on complex interactions between the orientation of the grating, aperture edges, and topographic structure of the visual map. Here, we aimed to test how aperture biases described during perception affect working memory (WM) decoding. For memoranda, we used gratings multiplied by radial and angular modulators to generate orthogonal aperture biases for identical orientations. Therefore, if WM representations are simply maintained sensory representations, they would have similar aperture biases. If they are abstractions of sensory features, they would be unbiased and the modulator would have no effect on orientation decoding. Neural patterns of delay period activity while maintaining the orientation of gratings with one modulator (e.g. radial) were interchangeable with patterns while maintaining gratings with the other modulator (e.g. angular) in visual and parietal cortex, suggesting that WM representations are insensitive to aperture biases during perception. Then, we visualized memory abstractions of stimuli using models of visual field map properties. Regardless of aperture biases, WM representations of both modulated gratings were recoded into a single oriented line. These results provide strong evidence that visual WM representations are abstractions of percepts, immune to perceptual aperture biases, and compel revisions of WM theory.

**\*For correspondence:**
clayton.curtis@nyu.edu

**Competing interest:** The authors declare that no competing interests exist.

## eLife assessment

This paper provides **valuable** insights into the neural substrates of human working memory. Through clever experimental design and rigorous analyses, the paper provides **compelling** evidence that the working memory representation of stimulus orientation is a reformatted version of the presented stimulus, though more work is needed to establish more generally that visual working memories are abstractions of percepts. This work will be of broad interest to cognitive neuroscientists working on the neural bases of visual perception and memory.

## Introduction

Following now classic studies demonstrating that fMRI patterns of voxel activity in human early visual cortex can be used to decode the contents of visual working memory (WM; *Harrison and Tong, 2009*; *Serences et al., 2009*), decoding WM content from visual cortex has been a workhorse for neuroimaging studies testing aspects of the sensory recruitment hypothesis of WM. This incredibly influential hypothesis posits that visual WM storage utilizes the encoding machinery in the visual cortex, assuming that memory and perception utilize similar mechanisms (*Postle, 2006*; *Curtis and D'Esposito, 2003*; *D'Esposito and Postle, 2015*; *Serences, 2016*).

Research has produced evidence for and against this hypothesis. On the one hand, WM representations can be decoded from the activity patterns as early as primary visual cortex (V1; *Harrison and Tong, 2009*; *Serences et al., 2009*; *Riggall and Postle, 2012*; *Sprague et al., 2014*; *Rahmati et al., 2018*; *Curtis and Sprague, 2021*). There is even some evidence that classifiers trained on

data collected from early visual cortex while participants are simply viewing stimuli (e.g. oriented gratings) can be used to decode the contents of WM (*Harrison and Tong, 2009*; *Rademaker et al., 2019*; *Albers et al., 2013*). The assumption here is that if sensory representations generated via bottom-up processing are interchangeable with WM representations, then the representation itself is perceptual in nature (although see *Lee et al., 2012*). Finally, the degree to which WM representations in early visual cortex are epiphenomenal or only support memory under impoverished laboratory conditions remains controversial. Some evidence suggests, however, that the neural circuitry in early visual cortex can simultaneously maintain WM representations while encoding incoming and potentially distracting percepts (*Hallenbeck et al., 2021*; *Rademaker et al., 2019*; *Lorenc et al., 2018*; *Iamshchinina et al., 2021*). Moreover, trialwise variations in these decoded WM representations predict key behavioral factors like errors (*Ester et al., 2013*) and uncertainty of memory (*Li et al., 2021*). Distractor-induced distortions in WM representations also predict the direction and degree of distractor-induced memory errors (*Hallenbeck et al., 2021*). Together, it appears as if memory-guided behaviors depend on a readout of these representations in early visual cortex.

On the other hand, several pieces of evidence are at odds with the sensory recruitment hypothesis of WM. With perhaps the exception of spatial WM (*Saber et al., 2015*; *Hallenbeck et al., 2021*; *Li and Curtis, 2023*; *van Kerkoerle et al., 2017*; *Supèr et al., 2001*), persistent activity, the most conclusive neural mechanism of WM, is not characteristic of V1 neurons (*Leavitt et al., 2017*; *Curtis and Sprague, 2021*). As mentioned above, fMRI patterns during perception can be used to predict WM content. However, decoding is usually worse compared to when WM data are used to train decoders (*Harrison and Tong, 2009*; *Rademaker et al., 2019*), especially in parietal cortex (*Albers et al., 2013*; *Rademaker et al., 2019*). WM representations in early visual cortex also appear to change over time from when encoding the memoranda to its maintenance throughout the retention interval. These changes appear to reflect reformatting of the representation from one that is more sensory-like to one during WM that is more connected to the demands of the memory-guided behavior (*Kwak and Curtis, 2022*; *Li and Curtis, 2023*; *Henderson et al., 2022*) and may explain how WM representations in V1 survive distraction (*Rademaker et al., 2019*; *Hallenbeck et al., 2021*).

Most of these studies that provide evidence for and against the sensory recruitment hypothesis of WM relied on decoding the orientation of gratings with fMRI patterns of voxel acitivty. The general linking hypothesis, therefore, assumes that successful orientation decoding depends on the unique patterns of activity originating from inhomogeneous sampling of orientation columns at fine scales across voxels (*Boynton, 2005*; *Haynes and Rees, 2005*; *Kamitani and Tong, 2005*). However, recent research suggests that coarse, not fine, scale biases at the retinotopic map level, such as a global preference for cardinal and radial orientations (*Freeman et al., 2011*; *Freeman et al., 2013*; *Mannion et al., 2010*; *Roth et al., 2022*) underlie decoding of orientation during perception. Rather than just a reflection of fine-scale sampling of orientation tuned neurons, orientation decoding also relies on complex interactions between the stimulus's orientation, its bounding aperture, and topographic inhomogeneities across the visual field map (*Carlson, 2014*; *Roth et al., 2018*). Despite changes to the hypothesis linking successful decoding of perceived orientation to its underlying causes, it remains unknown how WM decoding might be affected by these coarse-scale biases.

Here, we directly address this gap by testing how aperture biases affect WM decoding, as well as leveraging these carefully manipulated stimulus properties of gratings to test how sensory-like are WM codes. In order to disambiguate the contributions to orientation decoding, we used as memoranda stimuli with aperture biases that were either aligned with or orthogonal to a grating's orientation (*Roth et al., 2018*). Previewing our results, we found that WM but not perceptual representations in early visual cortex were immune to aperture biases. Using models of V1 (*Simoncelli et al., 1992*) and techniques to visualize the spatial patterns associated with seeing and remembering oriented gratings (*Kok and de Lange, 2014*; *Kwak and Curtis, 2022*; *Yoo et al., 2022*; *Zhou et al., 2022*; *Favila et al., 2022*), WM representations were recoded into line-like patterns across retinotopic cortex. Together, these findings provide strong evidence that visual WM representations are not sensory-like in nature. They are abstractions of percepts and provide evidence that compels revisions to the sensory recruitment hypothesis of WM.

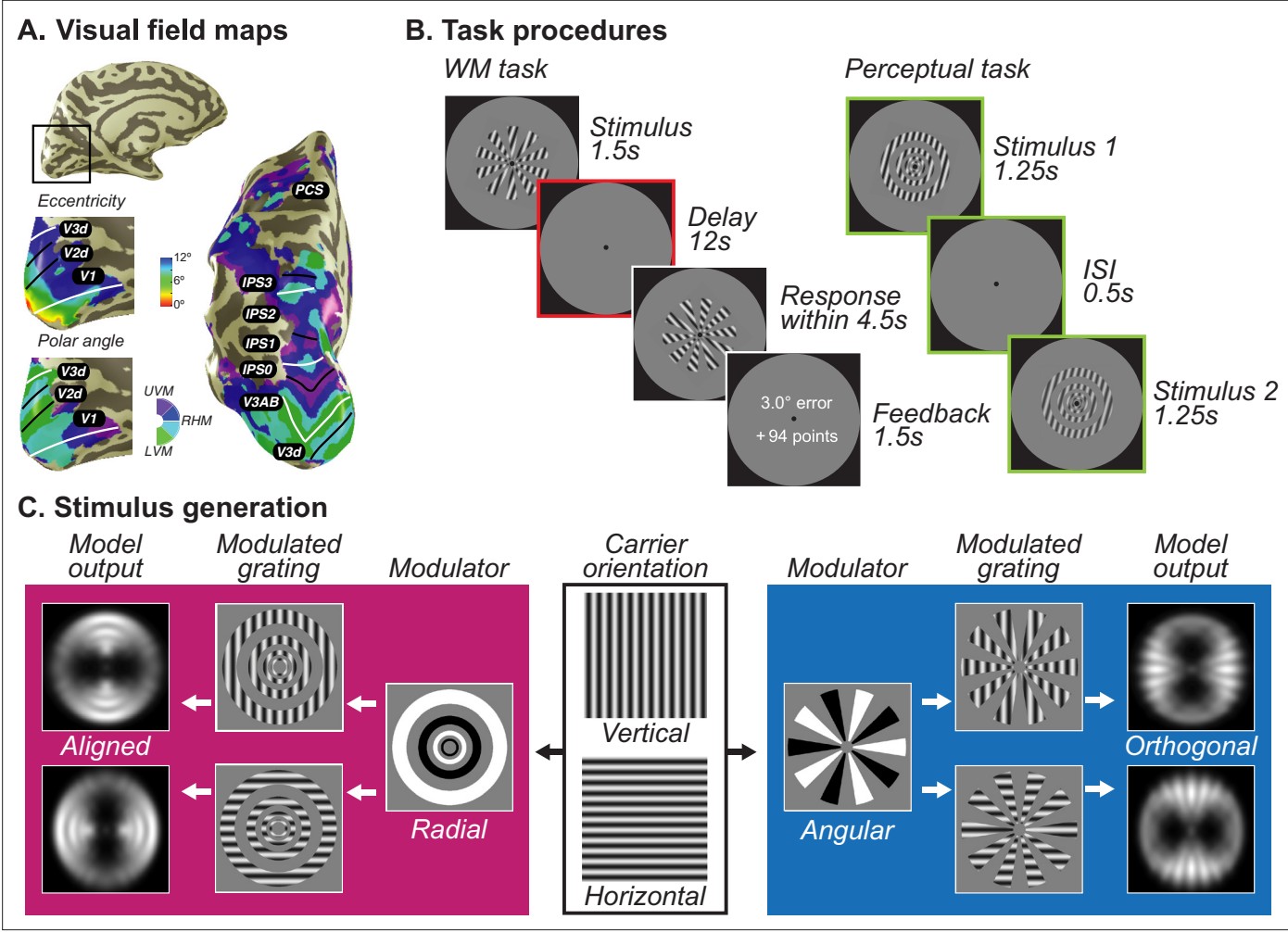

**Figure 1.** Population receptive field mapping, trial design, and stimuli generation schema. (**A**) A separate retinotopic mapping session was used to estimate voxel receptive field parameters for defining visual field maps in visual, parietal, and frontal cortices. Example participant's left hemisphere is shown. White lines denote the boundaries at the upper vertical meridian (UVM) and black lines denote the lower vertical meridian (LVM). (**B**) For the WM task (left), participants maintained the oriented stimuli over a 12 s retention interval and rotated a recall probe to match their memory. More points were awarded for less errors. For the perceptual control task (right), participants viewed the stimuli twice in a row with a short ISI and asked to decide which one has a higher contrast; it places no demand on remembering orientation. Colors denote different epoch of interests, green denotes stimulus epoch while red denotes delay epoch. (**C**) Each of the stimuli was created by multiplying a vertical or horizontal grating by a radial or angular modulator. These stimuli were used as input to the model. For radial modulated gratings (left in magenta), the model exhibits a radial preference: larger responses to vertical gratings along the vertical meridian and larger responses to horizontal gratings along the horizontal meridian. However, for angular modulated gratings (right in blue), the orientation preference is tangential: larger responses to vertical gratings along the horizontal meridian and larger responses to horizontal gratings along the vertical meridian. Here, we demonstrate the stimulus and aperture bias using vertical and horizontal carrier orientations. In the experiment, the carrier orientations were 15°, 75°, and 135° clockwise from vertical with random jitter (<7°).

## Results

### Angular and radial modulators impact orientation decoding during perception but not memory

We measured fMRI blood-oxygen-level-dependent (BOLD) activity in retinotopic visual field maps (*Figure 1A*) in humans when participants performed a delayed orientation WM task using gratings with two types of modulators (*Figure 1B*; WM). Participants also performed a separate perceptual control experiment using the same type of stimuli, but without a WM delay (*Figure 1B*; perception). Stimuli were created by multiplying oriented sinusoidal gratings (the carrier) with an angular or a radial polar grating (the modulator) to generate orthogonal aperture biases despite having the same orientation (*Roth et al., 2018*). Specifically, the radial modulator evokes a coarse-scale bias aligned

with the carrier orientation, while the angular modulator evokes a coarse-scale bias orthogonal to the carrier orientation (*Figure 1C*). We predicted that if the format of the memorized orientation is sensory-like in nature, decoding would conform with the aperture bias.

We first aimed to demonstrate that during a simple perception task without WM the radial and angular modulators induce different aperture biases that impact orientation decoding. As predicted, we replicated (*Roth et al., 2018*) that classifiers trained to decode the orientation of gratings altered by one type of modulator could only decode the orientation of gratings altered by the same type of modulator (*Figure 2A*, *Figure 2—figure supplement 1*; within). Classifiers could not cross-decode orientation gratings altered by the other type of modulator (*Figure 2A*, *Figure 2—figure supplement 1*; cross) presumably because classification depends on aperture biases that are orthogonal for radial and angular modulated gratings. Note that these effects were limited to visual field maps in early visual cortex (V1-V3).

Next, we focused on the patterns of late delay period activity during the WM task (*Figure 1B*) when the signals were temporally separated from those evoked during visual stimulation. We used this epoch of data for both training classifiers and testing decoding success. We first validated our methods by replicating successful orientation decoding in visual and parietal cortex separately for each type of modulator (*Figure 2B*, *Figure 2—figure supplement 2*; within; *Emrich et al., 2013*; *Ester et al., 2015*; *Harrison and Tong, 2009*; *Kwak and Curtis, 2022*; *Riggall and Postle, 2012*; *Sarma et al., 2016*; *Serences et al., 2009*; *Yu and Shim, 2017*). Turning to the critical test, we asked if a classifier trained on oriented gratings with one type of modulator (e.g. radial) could be used to successfully cross-decode gratings with the other type of modulator (e.g. angular). Indeed, we found that despite the orthogonal aperture biases induced by the two modulators, their patterns during WM maintenance were interchangeable. Within visual field maps in early and mid visual cortex (V1, V2, V3, V3AB), parietal cortex (IPS0/1, IPS2/3), and frontal cortex (sPCS), classifiers trained on different modulators could cross-decode the orientation of gratings (*Figure 2B*, *Figure 2—figure supplement 2*; cross). These results indicate that WM representations of orientation are immune to the aperture biases we demonstrated during perception.

To further test if WM representations are similar to perception, we next trained classifiers using data from the perceptual control task and measured the extent to which these classifiers can decode orientation during WM, and what effect the modulators have on decoding. In early visual cortex (V1-V3), we found that classifiers trained during perception can be used to decode orientation information in WM, but only when the aperture bias is aligned with the orientation of the grating (i.e. radial modulator; *Figure 2C*, *Figure 2—figure supplement 3*; within). Similarly, we only observed significant WM decoding across modulator types in early visual cortex when classifiers were trained during perception of the radial (aligned with orientation) but not angular (orthogonal to orientation) modulated grating (*Figure 2C*, *Figure 2—figure supplement 3*; cross). These results indicate that WM representations of orientation, which are not biased by aperture, are only similar to perceptual representations when they happen to align with the aperture biases induced during perception. Note that V3AB was a notable exception in that orientation could be decoded regardless of the type of modulator used for training or testing.

## WM representations are recoded into abstractions of percepts

The results thus far imply that WM representations in early visual cortex are distinct from perceptual representations. Moreover, WM representations are immune to the aperture biases during perception perhaps because they have been recoded into another format during memory. Next, we aimed to visualize changes in format during perception and WM for oriented gratings with orthogonal aperture biases. We hypothesized that participants recoded in WM the carrier orientation of gratings, regardless of the type of modulator, into line-like images encoded in the spatial distribution of response amplitudes across topographic maps (*Kwak and Curtis, 2022*; *Li et al., 2021*). Again using the late delay period activity during the WM task, we constructed the spatial profile of neural activity within visual field maps (*Kok and de Lange, 2014*; *Kwak and Curtis, 2022*; *Yoo et al., 2022*) for both radial and angular modulated orientation gratings (*Figure 3A*, *Figure 3—figure supplement 1*). Specifically, for each voxel, we weighted its receptive field (the exponent of a Gaussian distribution) by the delay period amplitude and then summed across all voxels within an ROI (see *Equation 1* in Methods). Then, we rotated the reconstruction map for each orientation such that they were all centered at zero

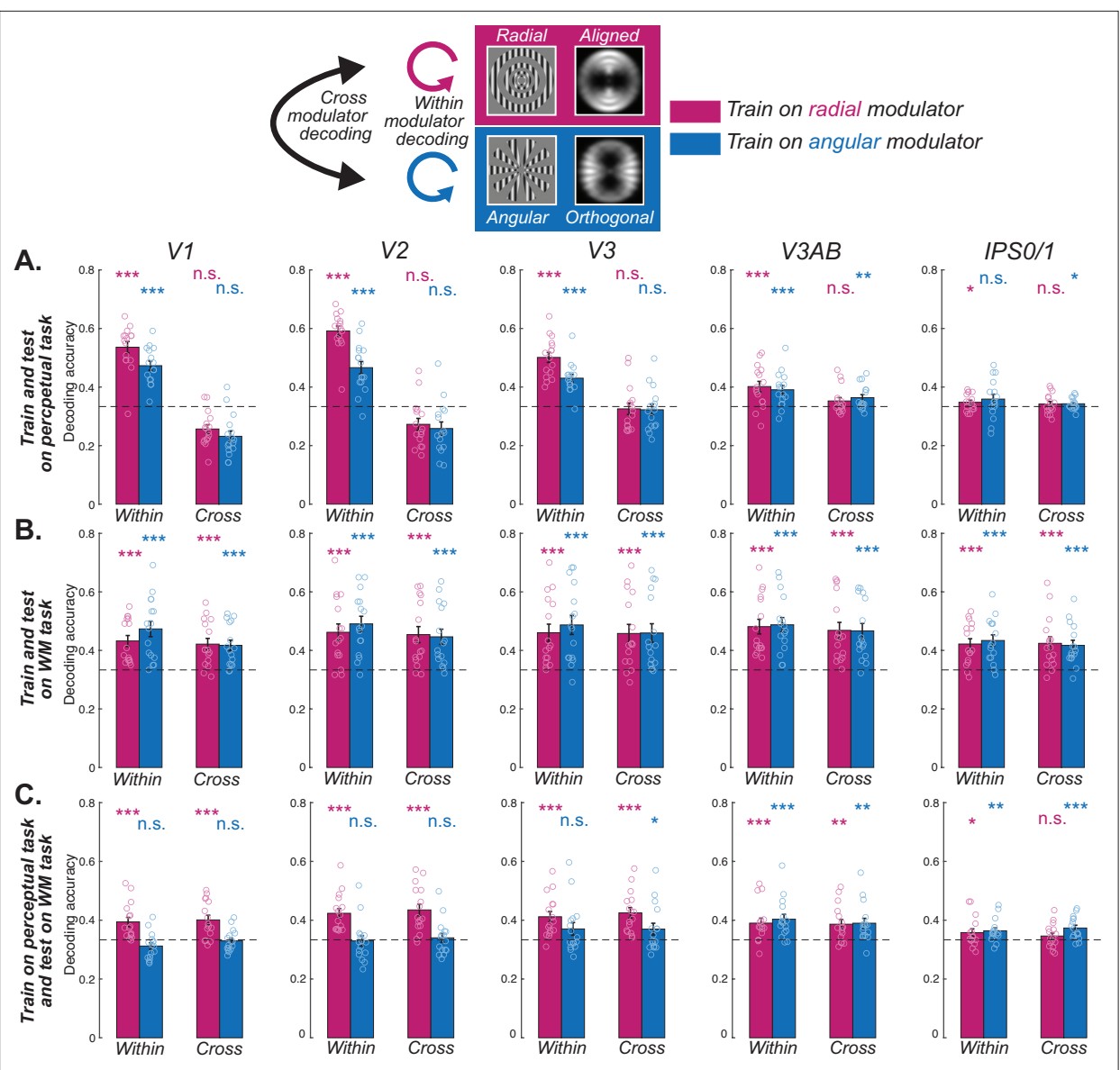

**Figure 2.** Decoding orientation during WM and perception. (**A**) Orientations could be decoded only within each kind of modulator, but not across different modulators in visual cortex, indicating the influence of the aperture bias on the stimulus in the perceptual task. (**B**) Orientations could be decoded both within and cross modulators in both visual and parietal cortices, suggesting a shared format during the late delay epoch in the WM task. (**C**) When training the classifier based on the neural pattern of the radial modulator (magenta) in the perceptual task, orientations of both radial (within) and angular (cross) modulators could be decoded during the WM late delay epoch in the visual cortex. However, training the classifier based on the angular modulator (blue) could not be generalized, except for V3AB. Results suggest that neural patterns during WM late delay are only similar to perceptual representations when their aperture bias aligns with the orientation bias (radial modulator) in early visual cortex (V1–V3). *p<0.05, **p<0.01, ***p<0.001, n.s. Not significant. Error bars represent ±1 SEM. Small circles for each bar represent individual data (n=16). Dashed horizontal line denotes theoretical chance level (1/3), but results are based on non-parametric permutation tests. Results for all ROIs can be seen in *Figure 2—figure supplements 1–3*. Statistical results can be seen in *Supplementary file 1a-c*.

The online version of this article includes the following figure supplement(s) for figure 2:

**Figure supplement 1.** Within and cross-modulator decoding results by using the stimulus period of the perceptual control task for all ROIs.

**Figure supplement 2.** Within and cross-modulator decoding results by using the late delay period of the WM task for all ROIs.

**Figure supplement 3.** Within and cross-modulator decoding results by training classifiers based on the stimulus period of the control task and testing them on the late delay period of the WM task for all ROIs.

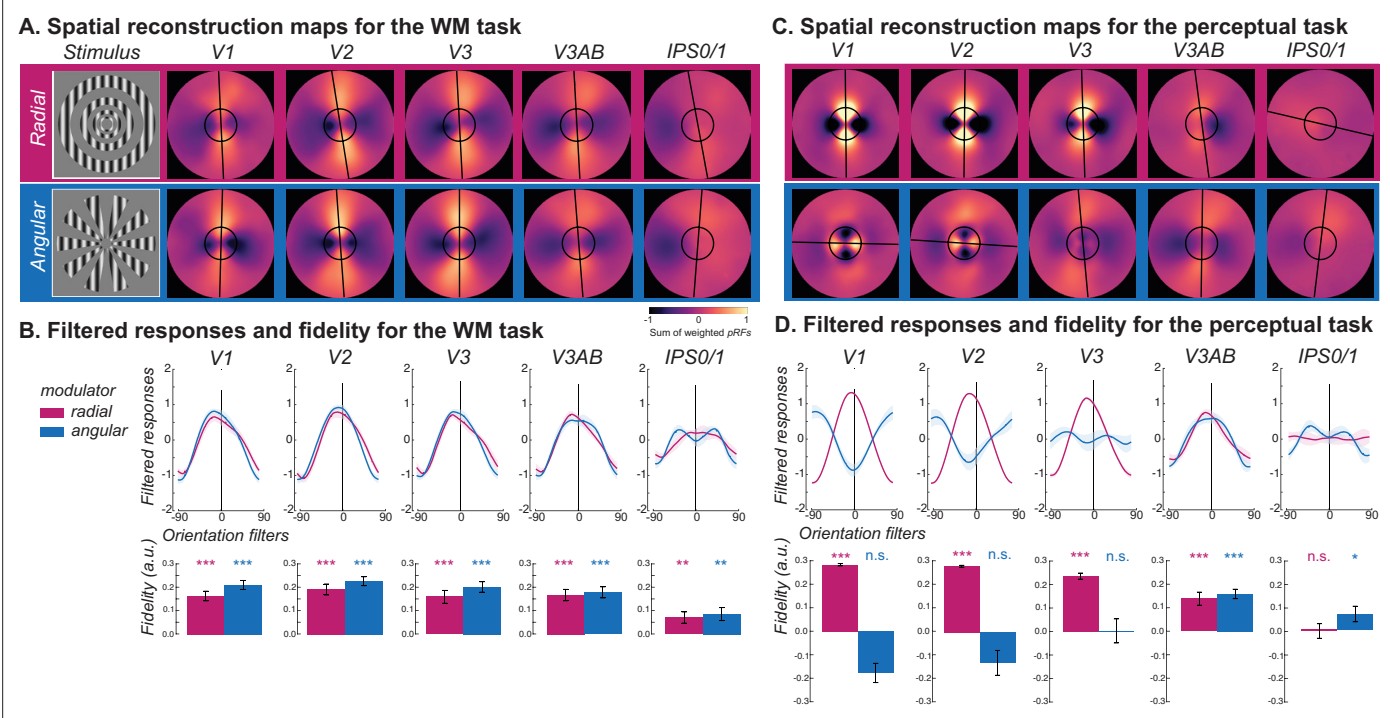

**Figure 3.** Visualizing WM and perception of radial and angular modulated oriented gratings. (**A**) Line-like patterns emerged across maps of visual space matching the memorized orientation of carrier gratings regardless of the type of modulator (radial - magenta; angular - blue) during the late delay period of the WM task. Spatial maps were rotated such that all orientations were aligned at 0° (top). The warmer colors correspond to increased amplitude of BOLD activity in voxels with receptive fields corresponding to that portion of the visual field. Best fitting lines (black lines) and the size of the stimulus (black circles) are overlaid. (**B**) Quantitative analysis confirmed the line-like patterns being aligned with the carrier orientation in the WM task. Filtered responses (top row) represent the sum of pixel values within the area of a line-shaped mask (12° length) oriented –90°–90°, where 0° represents the true orientation. Fidelity values (bottom row) are the result of projecting the filtered responses to 0° (see Methods), where higher fidelity values indicate stronger stimulus orientation representations. (**C**) Unlike the WM task, during the perception task the angle of the line-like patterns depended on the type of modulator in early visual areas (V1 and V2), where the line matched the orientation of the aperture bias, not the carrier. Note how the line is orthogonal to the angular modulated carrier in early visual cortex (V1 and V2) but not in later visual field maps (e.g. V3A/B). (**D**) During the perception task, the line-like representations in early visual cortex for radial but not angular modulated orientations result in strong filtered responses and fidelities. *p<0.05, **p<0.01, ***p<0.001. Error bars represent±1SEM (n=16). Results for all ROIs can be seen in Figure 3, *Figure 3—figure supplements 1 and 2*. Statistical results can be seen in *Supplementary file 1d-e*.

The online version of this article includes the following figure supplement(s) for figure 3:

**Figure supplement 1.** Spatial reconstruction results for the WM task across all ROIs.

**Figure supplement 2.** Spatial reconstruction results for the perceptual control task across all ROIs.

degrees (vertical meridian) and averaged across all orientation conditions. Clearly, the visualization technique confirmed our hypothesis and revealed a line encoded in the amplitudes of voxel activity at the angle matching the target orientation during the WM delay in V1-V3AB and IPS0/1 (*Figure 3A*), but not other ROIs (see *Figure 3—figure supplement 1* for details). Critically, these line-like representations were matched to the carrier orientation and not the aperture biases induced by the modulator. We statistically confirmed these effects by quantifying the fidelity of reconstructions of the carrier orientation (*Figure 3B*).

Next, we performed the same analyses using the data from the perception control experiment. The spatial maps in V1 and V2 revealed line-like representations of the gratings; however, they were aligned with the carrier orientation only when it was radial modulated (*Figure 3C/D*, for other ROIs see *Figure 3—figure supplement 2*); angular modulated gratings produced line-like representations that were orthogonal carrier orientation reflecting the influence of stimulus vignetting (*Roth et al., 2018*). We compared those reconstructed spatial maps with the simulated responses from an image-computable model based on the properties of V1 (*Roth et al., 2018*; *Simoncelli et al., 1992*). We simulated model outputs of both types of modulated gratings as well as line-like images

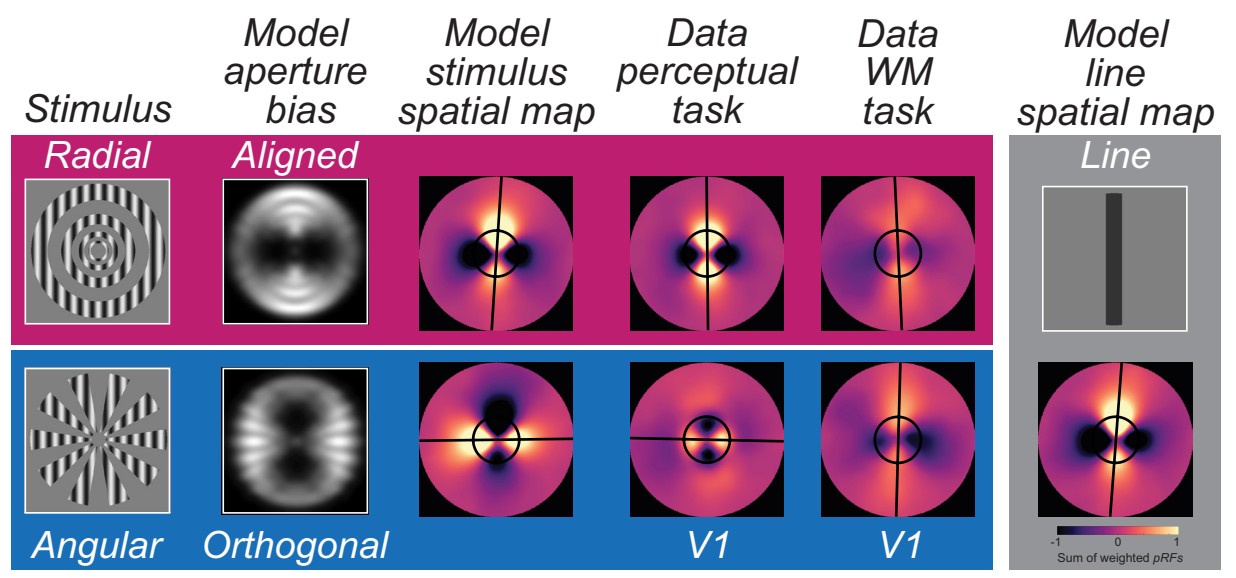

**Figure 4.** Modeling and reconstructing spatial maps of perceptual and mnemonic representations in V1. At the left, we illustrate the output of the model of V1 depicting the aperture biases aligned and orthogonal to the carrier orientation for radial and angular modulators, respectively. Using these modeled responses as inputs, we visualized the population code employing the measured pRF parameters from V1 (see Methods). In the modeled stimulus spatial map, line-like representations match the aperture biases, which in turn matches the observed data from V1 during the perception task. Critically, during WM storage, the line-like representations are aligned with the memorized carrier orientation in V1, regardless of modulator type. At the right and using the same model of V1, we visualize a WM representation in V1 assuming that participants are maintaining in WM a simple line that matches the carrier orientation.

at angles matching the orientation of the carrier grating (*Figure 4*). The spatial maps based on model responses matched those in the perception control task. They showed a clear orthogonal orientation bias induced by the stimulus aperture, while the results for line-like images matched the spatial profile of neural activity during the WM delay.

Overall, these results provide solid evidence that mnemonic representations are flexibly recoded into a spatial topographic format that is line-like in nature with angles matching the target orientation. WM appears immune to the aperture biases because its format is an abstraction of the perceptual features underlying the biases.

## Discussion

In attempts to adjudicate conflicting results between monkey and human studies of the role of the PFC in WM, *Curtis and D'Esposito, 2003* hypothesized that the PFC might be the source of top-down control signals that target neurons in sensory areas where WM representations are stored (see also *Postle, 2006*; *Curtis and Sprague, 2021*). Although just a speculation at the time, a few years later key evidence emerged. The orientations of memorized gratings could be decoded from the patterns of voxel activity during WM delays in primary visual cortex (*Harrison and Tong, 2009*; *Serences et al., 2009*), supporting the prediction that WM representations could be stored in sensory cortex. What became known as the *sensory recruitment hypothesis* of WM emerged shortly after (*Postle, 2006*; *Curtis and D'Esposito, 2003*; *D'Esposito and Postle, 2015*; *Serences, 2016*), which simply stated that the same neural encoding mechanisms used for perception are also utilized to store WM representations. The findings from the current study have two major and direct implications for this highly influential theory of WM. As we detail next, they provide conclusive evidence that neural representations of percepts are not the same as neural representations of memory, even in early visual cortex. Instead, our evidence indicates that WM representations are reformatted abstractions of percepts.

## Orientation decoding during perception and memory depends on distinct mechanisms

First, we situate our results within existing evidence that the neural mechanisms that support perception and WM are shared. The patterns of fMRI voxel activity in early visual cortex during perception of an oriented grating can be used to predict the orientation of a grating stored in WM (*Harrison and Tong, 2009*; *Rademaker et al., 2019*). Data such as these have been used to support the idea that the representation of WM features in early visual cortex are sensory-like in nature, presumably because orientation decoding during perception and WM both depend on the activities of neurons with orientation tuning (*Ester et al., 2013*; *Harrison and Tong, 2009*; *Serences et al., 2009*). However, the reason why patterns of voxel activity in early visual cortex can be used to decode orientation in simple perception studies has come under scrutiny. Initially, orientation decoding was thought to reflect random voxel sampling of the fine-scale columnar distributions of neurons with orientation tuning (*Boynton, 2005*; *Haynes and Rees, 2005*; *Kamitani and Tong, 2005*). Theoretical (*Carlson, 2014*) and empirical work (*Freeman et al., 2011*; *Freeman et al., 2013*; *Roth et al., 2018*) argued that decoding depended instead on coarse-scale factors. Specifically, it appears that orientation decoding relies to some degree on the complex interaction between a grating's orientation, its bounding aperture, and the non-isotropic distribution of orientation tuned neurons across the topographic map of V1. (*Roth et al., 2018*), using the same modulated orientation gratings we used, demonstrated that fMRI decoding of orientation depends on the coarse-scale aperture biases the modulators evoke. Here, we leveraged the precise control of these aperture biases evoked by the modulators to test if WM representations also depend on these aperture biases. First, we replicated the aperture biases in early visual cortex during perception reported by *Roth et al., 2018*; *Figure 2A*. Second and remarkably, we found that decoding orientation from patterns of activity in early visual cortex during WM delays were immune to the aperture biases noted during perception (*Figure 2B*). Third, when training classifiers based on the perceptual task, we could only decode orientation during WM when the aperture bias was aligned with the orientation of the carrier grating (*Figure 2C*). Together, these results provide strong and direct evidence that the patterns of neural activity during perception of an oriented grating are distinct from the patterns during WM for the same grating.

## Seeing is believing: WM representations are abstractions of percepts

Next, we addressed *how* WM representations of orientation are different from those during perception. To do so, we first visualized the spatial pattern of population activity within visual field maps by projecting voxel activity from cortex into spatial maps of activity in the coordinates of the physical screen within which stimuli were presented (*Kok and de Lange, 2014*; *Kwak and Curtis, 2022*; *Yoo et al., 2022*; *Li and Curtis, 2023*; *Favila et al., 2022*; *Zhou et al., 2022*). We found line-like representations of WM across many of the visual field maps in the dorsal stream whose angle matched the orientation of memorized gratings regardless of the modulator (radial, angular) and thus, regardless of the alignment between the carrier orientation and induced aperture bias (*Figure 3A*). During the perception control task, the line-like patterns were also present in the population response, but the angles of these lines matched the axis of the aperture bias rather than the grating's orientation (*Figure 3B*), again confirming differences between perception and memory. Finally, we used a computational model of V1 that simulated the aperture biases induced by the modulators (*Roth et al., 2018*; *Simoncelli et al., 1992*). Consistent with our empirical data, we found line-like stripes across retinotopic V1 aligned to the aperture bias and not the carrier orientation, providing a plausible explanation for why WM decoding depends on factors other than orientation. Instead, we propose that WM for oriented gratings, no matter what the aperture is, are reformatted into simple spatial codes, like a line (*Kwak and Curtis, 2022*; *Li and Curtis, 2023*). These line-like patterns are remarkably similar to simulations of a physical line input to the model of V1 (*Figure 4*), suggesting people are storing a simplified abstraction of the physical stimulus. These results also may explain why WM representations do not appear to undergo normalization like perceptual representations (*Bloem et al., 2018*).

## Concluding remarks

In summary, we found the WM decoding of orientation is immune to the aperture biases that drive decoding during perceptual studies of orientation. Moreover, WM representations are reformatted into efficient abstractions of percepts such that they most closely support memory guided behavior.

Although our previous study also found evidence that oriented gratings were recoded into line-like representations (**Kwak and Curtis, 2022**), here we demonstrate that those representations are not driven by aperture biases, but instead reflect abstract line-like representations. These results together necessitate revisions to the sensory recruitment hypothesis of WM because the same stimulus is supported by distinct, and not as predicted interchangeable, patterns of neural activity during perception and memory. If seeing and remembering a stimulus depends on the same encoding mechanisms then one would predict an interchangeable pattern. At the very least, the sensory recruitment hypothesis must be modified to take into account both how WM representations differ from perceptual representations, and how WM representations can morph into different formats that likely depend on the goal of the memory-guided behavior.

## Methods

### Subjects

Sixteen neurologically healthy volunteers (including the two authors; five females; 20–54 years old) with normal or corrected-to-normal vision participated in this study. Each participant completed three experimental sessions (two for the WM task and one for the control task, ~1 hr 30 min each) and one to two sessions of retinotopic mapping and anatomical scans (~2 hr). The experiments were conducted with the informed consent of each participant. The experimental protocols were approved by the University Committee on Activities involving Human Subjects at New York University associated with IRB-FY2017-1024.

### Stimuli

Stimuli were created by multiplying two gratings (a carrier and a modulator; **Figure 1C**; **Roth et al., 2018**). The carrier grating consisted of a large, oriented sinusoidal Cartesian grating (contrast = 0.8, spatial frequency = 1 cycle/°) presented within an annulus (inner diameter: 1.2°; outer diameter: 12°). The spatial phase of the carrier grating was either 0 or π, counterbalanced within each run. We generated 180 orientations for the carrier grating to cover the whole orientation space during the continuous report task. A gray circular aperture with a diameter of 24.8° (equal to the height of the screen) was presented as the background throughout the experiment.

The modulator grating was polar-transformed and square wave with hard edges, so that when multiplying with the carrier, it alternates the phase of the carrier and creates apertures. On half of the runs, the modulator produced a set of rings starting from the fovea (radial modulator, scaled with eccentricity). While on the other half of the runs, the modulator produced a set of inward-pointing wedges encircling the fovea (angular modulator). Importantly, the image-computable model of V1 (described below; **Roth et al., 2018**; **Simoncelli et al., 1992**) predicted a radial preference for the radial modulated gratings, but a tangential preference for the angular modulated gratings. Therefore, the radial modulator induced a bias that is consistent with the carrier orientation while the angular modulator induced a bias that is orthogonal to the carrier orientation. The modulator grating was either sine phase or cosine phase, counterbalanced within each run and orthogonal to the carrier's phase. The example of modulators in **Figure 1C** shows one kind of the phase conditions.

Importantly, when changing the orientation of the stimuli for each trial, it only changed the orientation of the carrier grating but not the modulator grating. Therefore, any fMRI activity measured could be attributed to either the orientation of the carrier grating, or an interaction between the orientation of the carrier grating and the static modulator grating.

### Apparatus setup

All stimuli were generated by using PsychToolBox in Matlab 2021b and presented by an LCD (VPixx ProPix) projector. The projected image spanned 36.2 cm in height and 64.4 cm in width. The spatial resolution is 1920 × 1080 for all tasks. The refresh rate is 120 Hz for the two tasks in the current study and 60 Hz for the retinotopic mapping tasks.

### fMRI task

Each participant completed two sessions for the WM task and one session for the control task on separate days. The two sessions for the WM task were acquired in 2 continuous days while having

several days intervals between the WM task and the control task to minimize the task confusion. The sequence of the two tasks was randomly assigned (10 subjects did the WM task first). For both tasks, each session consisted of 10 runs, which cost 1.5–2 hr. Each run had 12 trials for the WM task and 24 trials for the control task. Thus, participants completed 240 trials in total for both tasks. For each run, the target orientation (i.e. the carrier's orientation) were 15°, 75°, and 135° clockwise from vertical with random jitter (<7°).

## The WM task

Participants performed a delayed-estimation WM task where they need to report the remembered orientation for the target stimulus. Each trial began with 0.75 s of central fixation (subtended 0.8° diameters) followed by a target stimulus for 1.5 s. The stimulus was either radial modulated or angular modulated grating, presented in blocked designs and in interleaved order. After a 12 s delay period, participants were asked to rotate a recall probe with a dial to match the remembered orientation within a 4.5 s response window. To avoid visual afterimage, we inserted a 0.6 s noise mask at the beginning of the delay. The recall probe was the same type as the target stimulus to avoid forcing participants to represent the two stimulus types in an abstract manner. Again, when changing the orientation of the recall probe, only the carrier grating but not the modulator was changing. Participants were provided with feedback on the error they made and the points earned based on the error for each trial (100 points for 0°, no points for $\geq 50°$, 2 points for each degree). The feedback was displayed for 1.5 s and followed by an inter-trial-interval (ITI) of 6, 9, or 12 s.

## The perceptual control task

To better compare mnemonic formats with sensory representations, we asked participants to do an additional control task. Instead of asking participants to remember the orientation of the target stimulus, we presented it twice (1.25 s for each) with a short inter-stimulus-interval (ISI, 0.5 s) and asked participants to discriminate their contrast. The feedback was displayed for 0.5 s and followed by an inter-trial-interval (ITI) of 6, 9, or 12 s. Thus, the two target stimuli were exactly the same except for their contrast. The contrast for each stimulus was generated from a predefined set of 20 contrasts uniformly distributed between 0.5 and 1.0 (0.025 step size). We created 19 levels of task difficulty based on the contrast distance between the two stimuli. Thus, the difficulty ranged from choosing contrast pairs with the largest difference (0.5, easiest) to contrast pairs with the smallest difference (0.025, hardest). Task difficulty level changed based on an adaptive, 1-up-2-down staircase procedure (*Levitt, 1971*) to maintain performance at approximately 70% correct.

## Retinotopic mapping task

Each participant was scanned for a separate retinotopic mapping session (8–12 runs) to identify region-of-interest (ROI) and model each voxel's population receptive field (pRF). Participants ran in either type of attention-demanding tasks: random dot kinematogram (RDK) motion direction discrimination task (2 participants; *Mackey et al., 2017*) or an object image rapid serial visual presentation (RSVP) task (14 participants).

In the RDK motion discrimination task, participants maintained fixation at the center of the screen while covertly tracking a bar sweeping slowly but discretely across the screen in four directions (left-to-right, right-to-left, bottom-to-up, up-to-bottom). The bar was divided into three rectangular patches (one central patch and two flanking patches). The dot motion in one of the flanking patches matched the one in the central patch, while the other is the opposite. Participants were asked to discriminate which one is matched. The coherence of dot motions was 100% in the central patch, while the coherence in the flanking patches was staircase by using 2-up-1-down procedure to keep the task difficulty at about 75% accuracy (*Levitt, 1971*).

In the object image RSVP task, the moving bar that participants need to track consisted of six different object images. In each sweep, participants were asked to report whether the target object image existed among the six images by pressing a button. The target image was pseudo-randomly chosen for each run and was shown at the start of each run to help participants get familiar with it. The presentation duration of object bars was adjusted based on participants' accuracy in a staircase procedure.

## MRI data acquisition

MRI data were acquired on a Siemens Prisma 3T scanner with a 64-channel head/neck coil. For the WM task and the control task, BOLD contrast images were acquired using multiband (MB) 2D GE-EPI (MB factor of 4, 44 slices, 2.5x2.5 x 2.5mm voxel size, FoV 200x200 mm, TE/TR of 30/750ms, P → A phase encoding). Intermittently throughout each scanning session, we also acquired distortion mapping scans to measure field inhomogeneities with both forward and reverse phase encoding using a 2D SE-EPI readout and the number of slices matching that of the GE-EPI (TE/TR: 45.6/3537ms, 3 volumes per phase encode direction). BOLD contrast images for the retinotopic mapping task were acquired in a separate session with a higher resolution (MB factor of 4, 56 slices, 2x2 x 2mm voxel size, FoV 208x208 mm, TE/TR: 42/1300ms, P → A phase encoding). Similarly, we collected distortion mapping scans to measure field inhomogeneities with both forward and reverse phase encoding using a 2D SE-EPI readout and the number of slices matching that of the GE-EPI (TE/TR: 71.8/6690ms). Moreover, we also collected 2 or 3 T1 weighted (192 slices, 0.8x0.8 x 0.8mm voxel size, FoV 256x240 mm, TE/TR: 2.24/2400ms) and 1 or 2 T2 weighted (224 slices, 0.8x0.8 x 0.8mm voxel size, FoV 256x240 mm, TE/TR: 564/3200ms) whole-brain anatomical scans using the Siemens product MPRAGE for each participant.

## MRI data preprocessing

We used intensity-normalized high-resolution anatomical scans as input to Freesurfer's recon-all script (version 6.0) to identify pial and white matter surfaces, which were converted to the SUMA format. This anatomical image processed for each subject was the alignment target for all functional images. For functional preprocessing, we divided each functional session into two to six sub-sessions consisting of two to five task runs split by distortion runs (a pair of spin-echo images acquired in opposite phase encoding directions) and applied all preprocessing steps described below to each sub-session independently.

First, we corrected functional images for intensity inhomogeneity induced by the high-density receive coil by dividing all images by a smoothed bias field (15 mm FWHM), which was computed as the ratio of signal acquired with the head coil to that of the body coil. Then, to improve co-registration of functional data to the target T1 anatomical image, transformation matrices between functional and anatomical images were computed using distortion-corrected and averaged spin-echo images (distortion scans used to compute distortion fields restricted to the phase-encoding direction). Then, we used the distortion-correction procedure to undistort and motion-correct functional images. The next step was rendering functional data from native acquisition space into un-warped, motion-corrected, and co-registered anatomical space for each participant at the same voxel size as data acquisition (2.5 mm iso-tropic voxel). This volume-space data was projected onto the reconstructed cortical surface, which was projected back into the volume space for all analyses. Finally, we linearly detrended activation values from each voxel from each run. These values were then converted to percent signal change by dividing by the mean of the voxel's activation values over each run.

## Retinotopic mapping and region of interest (ROI) definition

Since the retinotopic mapping scans were acquired with a higher resolution than the experimental scans, we projected the retinotopic time series data onto the surface from its original space (2 mm), then from the surface to volume space at the task voxel resolution (2.5 mm). This ensured that estimates of variance-explained faithfully reflected the goodness of fit and were not impacted by smoothing incurred from transforming fit parameter values between different voxel grids.

We fitted a population receptive field (pRF) model with compressive spatial summation to the averaged time series across all retinotopy runs for each participant after smoothing on the surface with 5 mm FWHM Gaussian kernel (*Kay et al., 2013*; *Wandell et al., 2007*). Then, we projected the best-fit polar angle and eccentricity parameters onto each participant's inflated brain surface map via AFNI and SUMA. ROIs were drawn on the surface based on established criteria for polar angle reversals and foveal representations (*Mackey et al., 2017*; *Wandell et al., 2007*). We set a threshold to only include voxels with greater than 10% variance explained by the pRF model. We defined bilateral visual ROIs, V1, V2, V3, V3AB, IPS0, IPS1, IPS2, IPS3, iPCS, and sPCS.

## fMRI data analysis: decoding accuracy

All decoding analyses were performed using the multinomial logistic regression with custom code based on the Princeton MVPA toolbox (https://github.com/princetonuniversity/princeton-mvpa-toolbox; *PrincetonUniversity, 2016*). We used Softmax and cross entropy as the activation and performance functions, which are suitable for multi-class linear classification problems (*Kwak and Curtis, 2022*). The scaled conjugate gradient method was used to fit the weights and bias parameters.

### WM task decoding analysis

For the main task, we focused on the delay epoch to test the abstract representational format in WM. First, we performed within-modulator decoding from the same modulator type to verify the reliable orientation information during the WM delay epoch. Then, we conducted cross-modulator decoding from different modulator types (e.g. training on the angular modulator and testing on the radial modulator). We were mostly interested in the cross-modulator decoding results to examine whether WM forms an abstract representation across different modulator types.

Decoding analysis was performed on the beta coefficients acquired from running a voxel-wise general linear model (GLM) using AFNI 3dDeconvolve. For each participant, we used GLM to estimate the responses of each voxel to the stimulus encoding, delay, and response epochs. Note that, to better separate data from delay epoch from encoding epoch, we modeled the second half of the whole delay period (late delay). Using the whole delay did not change any of the results we reported here. Each epoch was modeled by the convolution of a canonical model of the hemodynamic impulse response function with a square wave (boxcar regressor) whose duration was equal to the duration of the corresponding epoch. Importantly, we estimated beta coefficients for every trial independently for the late delay epoch in performing the decoding analysis. Other epochs were estimated using a common regressor for all trials (*Rissman et al., 2004*). This method was used to capitalize on the trial-by-trial variability of the epoch of interest while preventing the trial-by-trial variability of other epochs from soaking up a large portion of variance which could potentially be explained by the epoch of interest. Six motion regressors were included to account for movement during the scan. Each voxel's beta coefficients were z-scored within each run independently before the decoding analysis.

We performed a three-way classification to decode the three target orientation conditions, which were 15°, 75°, and 135° clockwise from vertical. For within-modulator decoding, we used leave-one-run-out cross-validation procedure, in which all trials in one run were left out on each iteration to test the performance of the classifier trained on the data from all other runs. For cross-stimulus decoding, the classifier was trained on beta coefficients of all trials in one modulator condition and tested on all trials in the other modulator condition.

### Control task decoding analysis

To get better control and verify the existence of the stimulus vignetting effect (*Roth et al., 2018*), we conducted a purely perceptual task and performed the same analysis on the stimulus epoch data from this task. Based on previous findings, we expected to find reliable above-chance decoding performance for within-modulator decoding, but not for cross-modulator decoding.

### Cross-task decoding analysis

We also performed cross-task decoding to test how neural representational formats change for different task goals. For each modulator type (e.g. angular modulator), we trained the classifier based on the stimulus epoch data in the control task and tested it on both the stimulus epoch and the delay epoch data in the WM task for both modulator types (i.e. angular and radial modulators). We were mainly interested in testing the classifier on the late delay epoch data in the WM task. If the WM representations changed to a common format for both modulator types to match the orientation bias, we expected to find a reliable above-chance decoding when training the classifier based on the radial modulator but not the angular modular type. This is because the radial modulator induces a bias that is consistent with the carrier orientation, while the angular modulator induces an orthogonal bias compared to the carrier orientation.

## fMRI data analysis: spatial reconstruction

To visualize the spatial profile of neural activity during the epoch of interest, we projected voxel amplitudes onto the 2D visual field space for each orientation condition and each ROI across all participants. Specifically, we first averaged the beta coefficients (β) from GLM for all trials in each orientation condition. Then, for each voxel, we weighted its receptive field (the exponent of a Gaussian distribution) by the averaged β. Finally, we summed the weighted receptive fields across all voxels within a certain ROI for each orientation condition. To account for the individual differences in the pRF structure, we normalized the spatial profile for each participant and then got the averaged spatial profile across all participants. For generating all the spatial reconstruction maps, we downsampled the resolution of the visual field space such that each pixel corresponded to 0.1 of visual angle. Only voxels whose pRF eccentricities were within 20 degrees of visual angle were included in the reconstruction (*Kwak and Curtis, 2022*).

For each orientation condition $i$, the sum $S_i$ of all voxels' weighted receptive fields (assuming the number of voxels in a certain ROI is $m$) could be computed as *Equation 1*, where $j$ is the index of each voxel; $x_j$ , $y_j$ , and $\sigma_j$ are the center and width of the voxel's receptive field. x and y are the positions in the reconstruction map at which the receptive fields were evaluated.

$$S_i \;=\; \sum_{j=1}^{m} \beta_{j,i} \;\times\; e^{-\dfrac{\left(x_j - x\right)^2 + \left(y_j - y\right)^2}{2\sigma_j^2}} \tag{1}$$

To better visualize the line format, we fitted a first-degree polynomial to the reconstructed map in *Figure 3A and C* (black lines). Specifically, we selected pixels within the stimulus size with the top 10% image intensity and fit these pixels' coordinates to a first-degree polynomial with a constraint that the fitted polynomial passed through the center. To account for the difference in image intensity between different pixels, we conducted a weighted fit, in which the weight corresponds to the voxel's rank in terms of its image intensity. We were mainly interested in comparing the spatial reconstruction maps between the delay epoch in the WM task and the stimulus epoch in the control task. The visualization provided us with an intuitive understanding of how representational formats changed from perception to WM, and what drove the different decoding results.

## Model simulation: image-computable model of V1

We used an image-computable model to predict fMRI responses of V1 to different types of stimuli for visual perception (*Roth et al., 2018*). We first simulated model outputs to different modulator types and then predicted fMRI responses by using pRF sampling analysis. To better visualize the model predictions, we conducted the same spatial reconstruction based on the simulated fMRI responses.

## Simulate model outputs

The image-computable model was based on the steerable pyramid model of V1 (*Simoncelli et al., 1992*), a subband image transform that decomposes an image into orientation and spatial frequency channels. Responses of many linear receptive fields (RFs) were simulated, each of which computed a weighted sum of the stimulus image. The weights determined the spatial frequency and orientation tuning of the linear RFs, which were hypothetical basis sets of spatial frequency and orientation tuning curves of V1. RFs with the same orientation and spatial frequency tuning but different location preferences were channels. In the model, the number of spatial frequency channels, orientation channels, and orientation bandwidth were adjustable. For the model simulation, we used six orientation bands (bandwidth = 180°/6=30°) and a spatial frequency bandwidth of 0.5 octaves as in previous studies (*Kwak and Curtis, 2022*; *Roth et al., 2018*). Using four or more bands with correspondingly broader or narrower tuning curves yielded similar results supporting the same conclusions. The number of spatial frequency channels was determined by the size of the input image and the spatial frequency bandwidth. We used images that were 1920 × 1080 pixels, which resulted in 16 levels/scales for the model. The input images had the same configurations (size of fixation, inner aperture, outer aperture, etc) as the stimuli in both the WM task and the control task. The model outputs were images of the same resolution as the input images, in which each pixel can be thought of as a simulated neuron in

the retinotopic map of V1. Importantly, we summed the model responses across all orientation channels, which resulted in a model without any orientation tuning.

For both types of stimuli, we used three target orientations (15°, 75°, and 135° clockwise from vertical), two phases for the carrier (0 or π), and two phases for the modulator (sine or cosine phase). We first generated the model responses to each phase condition separately, then averaged them across all phases for each orientation condition. This yielded three sets of simulated voxel maps, within which we had 16 maps for all subbands. For the final predicted responses, we chose the subband with maximal responses (the 9th level), which corresponds to the spatial frequency of the stimulus (*Roth et al., 2018*).

## pRF sampling analysis

To simulate an fMRI voxel's response to the stimuli, each participant's pRF Gaussian parameters of V1 were used to weight the model outputs, which resulted in a weighted sum of neural responses corresponding to pRFs. For each orientation condition $i$, the sampled fMRI BOLD signal ($B_{i,j}$) for voxel $j$ with a pRF centered at $x_j$, $y_j$ and standard deviation of $\sigma_j$, is computed as the dot product between the pRF and the model output ($M_i$) as in *Equation 2*. x and y are the positions in the model outputs at which the receptive fields were evaluated.

$$B_{i,j} = \sum_{x,y} M_i \times e^{-\frac{(x_j - x)^2 + (y_j - y)^2}{2\sigma_j^2}} \tag{2}$$

Finally, we performed the same spatial reconstruction analysis on these simulated BOLD signals after normalizing (z-score) across the three orientation conditions. To account for the individual differences in the pRF structure, we normalized the spatial profile for each participant's simulation and then got the averaged spatial profile across all participants. This was done separately for each of the two modulators.

## Eye-tracking setup and analyses

For all imaging sessions, we measured eye position using an EyeLink 1000 Plus infrared video-based eye tracker (SR Research) mounted beneath the screen inside the scanner bore operating at 500 Hz. The camera always tracked the participant's right eye, and we calibrated using either a 9-point (WM task and perceptual control task) or 5-point (retinotopic mapping task) calibration routine at the beginning of the session and as necessary between runs. We monitored gaze data and adjusted pupil/corneal reflection detection parameters as necessary during and/or between each run.

We preprocessed raw gaze data using fully-automated procedures implemented within iEye_ts (https://github.com/clayspacelab/iEye, copy archived at *clayspacelab, 2024*). Eye positions were not monitored for S04 during the first and for S16 during both of the two WM task sessions due to technical issues. Overall, 97.72% (radial) and 96.79% (angular) of the total number of eye position sample points during the delay epoch of the WM task across all subjects were within 2° eccentricity from the center (the fixation and the stimulus subtended 0.8° and 12° diameter, respectively). The circular correlation between the polar angle of the target orientation and the polar angle of the eye positions was not significant for both the radial (mean = 0.030, s.d.=0.103, $t(14)=1.141$, p=0.273) and the angular (mean = 0.001, s.d.=0.099, $t(14)=0.056$, p=0.956) modulator, suggesting that the eye movements could not account for our findings.

## Quantification and statistical analysis

No data were excluded for analysis. All statistical results reported here were based on permutation tests over 1000 iterations. To test whether decoding accuracy was significantly greater than chance level (1/3), we generated permuted null distributions of decoding accuracy values for each participant, ROI, decoding type (within/cross), modulator type (angular/radial), and each time point for the temporal decoding analysis. On each iteration, we shuffled the training data matrix (voxels x trials) for both dimensions so that both voxel information and orientation labels were shuffled. Then, we performed the decoding analysis based on the shuffled data. This procedure was conducted for each of the 16 participants, resulting in 16 null distributions of decoding accuracy. Combining the null

decoding accuracy across all participants resulted in one t-statistic per permutation. To test across-participants decoding accuracy against chance level (1/3), we compared the t-statistic calculated from the intact data against the permuted null distribution of t-statistic for each condition and ROI. The p-value was calculated as the proportion of permuted t-statistics that were greater than or equal to the t-statistic using the intact data.

For the spatial reconstruction analysis, we computed reconstruction fidelity to quantify the amount of orientation information in each reconstruction map. Specifically, we first created line filters, whose length was equal to the stimulus's diameter (12°), with orientations evenly spaced between –90° and 90° in steps of 1°. Then, we created masks around these line filters based on two rules. First, coordinates formed an acute angle to the oriented line filter (dot product >0). Second, to constrain the width of line filters, the projected distance squared was less than 1000 (**Kwak and Curtis, 2022**), using different thresholds did not change the results. We chose pixels within these masked areas and summed up the intensities. After z-scoring the summed intensities within each orientation condition, we rotated the response function so that the center is the target orientation. The final tuning curve-like response function was averaged across all three orientation conditions. To compute fidelity, we projected the filtered responses at each orientation filter onto a vector centered on the true orientation (0°) and took the mean of all the projected vectors. Conceptually, this metric measured whether and how strongly reconstruction on average points in the correct direction.

The same procedure for statistical analysis was used for the reconstruction fidelity, with the exception that the null hypothesis for the t-statistic was 0. Specifically, the data-derived fidelity value was compared against the distribution of null fidelity values from shuffled data. To generate the null distribution, the matrix of beta coefficients was shuffled across both the voxel and orientation condition label dimensions, and the shuffled beta coefficients were used to weight the voxels' pRF parameters.

To test whether there were differences in decoding accuracy (and reconstruction fidelity value) between the decoding type and modulator type within each ROI, we used permutation-based two-way repeated-measures analysis of variance (ANOVA). For each permutation, we shuffled the condition labels (decoding type and modulator type) per participant and calculated the null F-statistic. We repeated this procedure 1000 times and got the null distribution of the F-statistic. We compared the F-statistic derived from the intact data with the null distribution to get the p-value. Significant effects were followed up with post-hoc paired-sample t-tests, and the p-value was calculated by comparing the t-statistic derived by the intact data against a permuted null distribution of t-statistics generated by shuffling condition labels. The p-value was corrected by using a false-discovery rate (FDR) procedure for multiple comparisons.

## Acknowledgements

This work was supported by National Institutes of Health Grants R01 EY016407 and EY033925 to CEC. We thank Jonathan Winawer for helpful comments on earlier versions of the manuscript, and NYU's Center for Brain Imaging for support.

## Additional information

### Funding

| Funder | Grant reference number | Author |
| --- | --- | --- |
| National Eye Institute | EY016407 | Clayton E Curtis |
| National Eye Institute | EY033925 | Clayton E Curtis |

The funders had no role in study design, data collection and interpretation, or the decision to submit the work for publication.

### Author contributions

Ziyi Duan, Data curation, Formal analysis, Validation, Investigation, Visualization, Writing - original draft, Writing – review and editing; Clayton E Curtis, Conceptualization, Resources, Software,

Supervision, Funding acquisition, Investigation, Visualization, Methodology, Project administration, Writing – review and editing

## Author ORCIDs
Ziyi Duan http://orcid.org/0000-0001-7567-4120
Clayton E Curtis https://orcid.org/0000-0003-0702-1499

## Ethics

Human subjects: The experiments were conducted with the informed consent of each participant. The experimental protocols were approved by the University Committee on Activities involving Human Subjects at New York University.

Reviewer #1 (Public Review): https://doi.org/10.7554/eLife.94191.3.sa1
Reviewer #2 (Public Review): https://doi.org/10.7554/eLife.94191.3.sa2
Author response https://doi.org/10.7554/eLife.94191.3.sa3

---

# Additional files

## Supplementary files

• Supplementary file 1. Decoding accuracy tables with statistical results. (**a**) Decoding accuracy for the stimulus-presenting epoch in the perceptual control task. (**b**) Decoding accuracy for the late delay epoch in the WM task. (**c**) Decoding accuracy for the cross-task decoding by training the classifier in the perceptual control task and testing it in the WM task. (**d**) Reconstruction fidelity values for the late delay epoch in the WM task. (**e**) Reconstruction fidelity values for the stimulus-presenting epoch in the perceptual control task.

• MDAR checklist

## Data availability

The processed fMRI data and raw behavioral data generated in this study have been deposited in the Open Science Framework at https://doi.org/10.17605/OSF.IO/KWS9B. Processed fMRI data contains extracted time series from each voxel of each ROI. We also make publicly available all code that was used to analyze the fMRI data, implement the theoretic model of V1, and generate the stimuli.

The following dataset was generated:

| Author(s) | Year | Dataset title | Dataset URL | Database and Identifier |
| --- | --- | --- | --- | --- |
| Duan Z, Curtis CE | 2024 | Data and code for "Visual working memories are abstractions of percepts" | https://doi.org/10.17605/OSF.IO/KWS9B | Open Science Framework, 10.17605/OSF.IO/KWS9B |

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
