## [Editor Report · eLife assessment]

This paper provides **valuable** insights into the neural substrates of human working memory. Through clever experimental design and rigorous analyses, the paper provides **compelling** evidence that the working memory representation of stimulus orientation is a reformatted version of the presented stimulus, though more work is needed to establish more generally that visual working memories are abstractions of percepts. This work will be of broad interest to cognitive neuroscientists working on the neural bases of visual perception and memory.

---

## [Referee Report · Reviewer #1 (Public Review)]

Summary:

The authors aim to test the sensory recruitment theory of visual memory, which assumes that visual sensory areas are recruited for working memory and that these sensory areas represent visual memories in a similar fashion to how perceptual inputs are represented. To test the overlap between working memory (WM) and perception, the authors use coarse stimulus (aperture) biases that are known to account for (some) orientation decoding in visual cortex (i.e., stimulus energy is higher for parts of an image where a grating orientation is perpendicular to an aperture edge, and stimulus energy drives decoding). Specifically, the authors show gratings (with a given "carrier" orientation) behind two different apertures: One is a radial modulator (with maximal energy aligned with the carrier orientation) and the other an angular modulator (with maximal energy orthogonal to the carrier orientation). When subject detect contrast changes in these stimuli (the perceptual task), orientation decoding only works when training and testing within each modulator, but not across modulators, showing the impact of stimulus energy on decoding performance. Instead, when subjects remember the orientation over a 12s delay, orientation decoding works irrespective of the modulator used. The authors conclude that representations during WM are therefore not "sensory-like", given that they are immune to aperture biases. This invalidates the sensory recruitment hypothesis, or at least the part assuming that when sensory areas that are recruited during WM, they are recruited in a manner that resembles how these areas are used during perception.

Strengths:

Duan and Curtis very convincingly show that aperture effects that are present during perception, do not appear to be present during the working memory delay. Especially when the debate about "why can we decode orientations from human visual cortex" was in full swing, many may have quietly assumed this to be true (e.g., "the memory delay has no stimuli, and ergo no stimulus aperture effects"), but it is definitely not self-evident and nobody ever thought to test it directly until now. In addition to the clear absence of aperture effects during the delay, Duan and Curtis also show that when stimulus energy aligns with the carrier orientation, cross-generalization between perception and memory does work (which could explain why perception-to-memory cross decoding also works). All in all, this is a clever manipulation, and I'm glad someone did it, and did it well.

Weaknesses:

There seems to be a major possible confound that prohibits strong conclusions about "abstractions" into "line-like" representation, which is spatial attention. What if subjects simply attend the end points of the carrier grating, or attend to the edge of the screen where the carrier orientation "intersects" in order to do the task? This may also result in reconstructions that have higher bold at areas close to the stimulus/screen edges along the carrier orientation. The question then would be if this is truly an "abstracted representation", or if subjects are merely using spatial attention to do the task.

Alternatively (and this reaches back to the "fine vs coarse" debate), another argument could be that during memory, what we are decoding is indeed fine-scale inhomogenous sampling of orientation preferences across many voxels. This is clearly not the most convincing argument, as the spatial reconstructions (e.g., Figure 3A and C) show higher BOLD for voxels with receptive fields that are aligned to the remembered orientation (which is in itself a form of coarse scale bias), but could still play a role.

To conclude that the spatial reconstruction from the data indeed comes from a line-like representation, you'd need to generate modeled reconstructions of all possible stimuli and representations. Yes, Figure 4 shows that a line results in a modeled spatial map that resembles the WM data, but many other stimuli might too, and some may better match the data. For example, the alternative hypothesis (attention to grating endpoints) may very well lead to a very comparable model output to the one from a line. But testing this would not suffice, as there may be an inherent inverse problem (with multiple stimuli that can lead to the same visual field model).

The main conclusion, and title of the paper, that visual working memories are abstractions of percepts, is therefore not supported. Subjects could be using spatial attention, for example. Furthermore, even if it is true that gratings are abstracted into lines, this form of abstraction would not generalize to any non-spatial feature (e.g., color cannot become a line, contrast cannot become a line, etc.), which means it has limited explanatory power.

Additional context:

The working memory and perception tasks are rather different. In this case, the perception task does not require the subject to process the carrier orientation (which is largely occluded, and possibly not that obvious without paying attention to it), but attention is paid to contrast. In this scenario, stimulus energy may dominate the signal. In the WM task, subjects have to work out what orientation is shown to do the task. Given that the sensory stimulus in both tasks is brief (1.5s during memory encoding, and 2.5s total in the perceptual task), it would be interesting to look at decoding (and reconstructions) for the WM stimulus epoch. If abstraction (into a line) happens in working memory, then this perceptual part of the task should still be susceptible to aperture biases. It allows the authors to show that it is indeed during memory (and not merely the task or attentional state of the subject) that abstraction occurs.

What's also interesting is what happens in the passive perceptual condition, and the fact that spatial reconstructions for areas beyond V1 and V2 (i.e., V3, V3AB, and IPS0-1) align with (implied) grating endpoints, even when an angular modulator is used (Figure 3C). Are these areas also "abstracting" the stimulus (in a line-like format)?

Review after revision:

(1) It's nice of the authors to simulate how a dot stimulus affects the image computable model, but this does not entirely address my concern about attention to endpoints. The assumption that attention can be used in the same manner as a physical stimulus to calculate stimulus energy is questionable. (also, why would a dot at 15º lead to high stimulus energy tangential to that orientation?). This simulation also does not at all address my concern about model mimicry (many possible inputs can lead to a line-like output).

(2) It's also nice that the authors agree that much more work needs to be done, and these results may not generalize to all forms of memory. Given this agreement, and until that "more work" is done, I strongly believe we should refrain from making hyperbolic claims that might preemptively imply all visual working memories are abstractions of percepts. Time (and much more work) will likely show things to be much more subtle and complex.

The work presented in this paper is cool, but it uses a specific case: spatial stimuli (gratings) with the task to remember orientation. This limits possible conclusions for several reasons (1) These results are specific to EVC, as visual maps are a prerequisite meaning that these results will not hold up in other, non-retinotopic areas. (2) The fact that subjects are "focusing" along the main stimulus axis (attention or not) can simply be a strategy employed by the majority of (but not all) subjects - a strategy that may not be necessary to do the task, and therefore not a canonical method of Abstraction. It may be a "shared preferred strategy" or something. (3) If subjects had to (for example) remember contrast, and not orientation, results may have been entirely different (I would hypothesize there is no line-like abstraction in this case). Vice versa, if the perceptual task would have been on orientation (instead of contrast), the authors admit that "participants would reformat the grating into a line-like representation to make the judgments" (quote from author's response under "Additional context"). Thus, the results may be entirely about the task/ cognitive state, and not about how perceptual information is abstracted into memory.

Instead of unveiling *the* working memory Abstraction, this work (very nicely) shows a specific instance of possible abstraction. A more correct (but admittedly, less "sexy") conclusion may be "Visual working memories of orientation can be abstracted into a line in early visual cortex". As it stands, the authors still do not acknowledge any of the alternatives that myself (see above) and the other reviewers have put forth, nor do they acknowledge recent work by Chunharas et al. (2023, BioRxiv), that directly applies principles of efficient coding to address the exact same question of working memory abstraction. The link between a "line-like" representation and efficient coding implied by the authors (in their response) is merely tentative to me, but it would be great if the authors could explain this further.

These were, and remain, the major weaknesses in the original submission, that in my view have not been adequately addressed by the authors, as many overly broad conclusions about abstractions are currently still present in the manuscript (in for example the title).

---

## [Referee Report · Reviewer #2 (Public Review)]

Summary:

In this work, Duan and Curtis addressed an important issue related to the nature of working memory representations. This work is motivated by findings illustrating that orientation decoding performance for perceptual representations can be biased by the stimulus aperture (modulator). Here, the authors examined whether the decoding performance for working memory representations is similarly influenced by these aperture biases. The results provide convincing evidence that working memory representations have a different representational structure, as the decoding performance was not influenced by the type of stimulus aperture.

Strengths:

The strength of this work lies in the direct comparison of decoding performance for perceptual representations with working memory representations. The authors take well-motivated approach and illustrate that perceptual and working memory representations do not share a similar representational structure. The authors test a clear question, with a rigorous approach and provide compelling evidence. First, the presented oriented stimuli are carefully manipulated to create orthogonal biases introduced by the stimulus aperture (radial or angular modulator), regardless of the stimulus carrier orientation. Second, the authors implement advanced methods to decode the orientation information, in visual and parietal cortical regions, when directly perceiving or holding an oriented stimulus in memory. The data illustrates that working memory decoding is not influenced by the type of aperture, while this is the case in perception. In sum, the main claims are important and shed light on the nature of working memory representations.

Weaknesses:

After the authors revised the original manuscript, a few of my initial concerns remain.

(1) Theoretical framing in the introduction. The introduction proposes that decoding of orientation information during perception does not reflect orientation selectivity, and it is instead driven by coarse scale biases. This is an overstatement. Recent work shows that orientation decoding is indeed influenced by coarse biases, but also reflects orientation selectivity (Roth, Kay & Merriam, 2022).

(2) The description of the image computable V1 model remains incomplete. The steerable pyramid is a model that simulates the responses of V1 neurons. To do so, it incorporates a set of linear receptive fields with varying orientation and spatial frequency tuning. However, the information that is lacking in the Methods is whether the implemented pyramid also included two quadrature phase pairs (odd and even phase Gabor filters making the output phase invariant). The sum of the squares of the responses to these offset phase filters computes the stimulus energy within each orientation and spatial frequency channel. Without this description, it is unclear what the model output represents.

---

## [Author Response]

The following is the authors’ response to the original reviews.

**Public Reviews:**

**Reviewer #1:**
Summary:The authors aim to test the sensory recruitment theory of visual memory, which assumes that visual sensory areas are recruited for working memory, and that these sensory areas represent visual memories in a similar fashion to how perceptual inputs are represented. To test the overlap between working memory (WM) and perception, the authors use coarse stimulus (aperture) biases that are known to account for (some) orientation decoding in the visual cortex (i.e., stimulus energy is higher for parts of an image where a grating orientation is perpendicular to an aperture edge, and stimulus energy drives decoding). Specifically, the authors show gratings (with a given "carrier" orientation) behind two different apertures: one is a radial modulator (with maximal energy aligned with the carrier orientation) and the other an angular modulator (with maximal energy orthogonal to the carrier orientation). When the subject detects contrast changes in these stimuli (the perceptual task), orientation decoding only works when training and testing within each modulator, but not across modulators, showing the impact of stimulus energy on decoding performance. Instead, when subjects remember the orientation over a 12s delay, orientation decoding works irrespective of the modulator used. The authors conclude that representations during WM are therefore not "sensory-like", given that they are immune to aperture biases. This invalidates the sensory recruitment hypothesis, or at least the part assuming that when sensory areas are recruited during WM, they are recruited in a manner that resembles how these areas are used during perception.Strengths:Duan and Curtis very convincingly show that aperture effects that are present during perception, do not appear to be present during the working memory delay. Especially when the debate about "why can we decode orientations from human visual cortex" was in full swing, many may have quietly assumed this to be true (e.g., "the memory delay has no stimuli, and ergo no stimulus aperture effects"), but it is definitely not self-evident and nobody ever thought to test it directly until now. In addition to the clear absence of aperture effects during the delay, Duan and Curtis also show that when stimulus energy aligns with the carrier orientation, cross-generalization between perception and memory does work (which could explain why perception-to-memory cross-decoding also works). All in all, this is a clever manipulation, and I'm glad someone did it, and did it well.Weaknesses:There seems to be a major possible confound that prohibits strong conclusions about "abstractions" into "line-like" representation, which is spatial attention. What if subjects simply attend the endpoints of the carrier grating, or attend to the edge of the screen where the carrier orientation "intersects" in order to do the task? This may also result in reconstructions that have higher bold at areas close to the stimulus/screen edges along the carrier orientation. The question then would be if this is truly an "abstracted representation", or if subjects are merely using spatial attention to do the task.Alternatively (and this reaches back to the "fine vs coarse" debate), another argument could be that during memory, what we are decoding is indeed fine-scale inhomogenous sampling of orientation preferences across many voxels. This is clearly not the most convincing argument, as the spatial reconstructions (e.g., Figure 3A and C) show higher BOLD for voxels with receptive fields that are aligned to the remembered orientation (which is in itself a form of coarse-scale bias), but could still play a role.To conclude that the spatial reconstruction from the data indeed comes from a line-like representation, you'd need to generate modeled reconstructions of all possible stimuli and representations. Yes, Figure 4 shows that line results in a modeled spatial map that resembles the WM data, but many other stimuli might too, and some may better match the data. For example, the alternative hypothesis (attention to grating endpoints) may very well lead to a very comparable model output to the one from a line. However testing this would not suffice, as there may be an inherent inverse problem (with multiple stimuli that can lead to the same visual field model).The main conclusion, and title of the paper, that visual working memories are abstractions of percepts, is therefore not supported. Subjects could be using spatial attention, for example. Furthermore, even if it is true that gratings are abstracted into lines, this form of abstraction would not generalize to any non-spatial feature (e.g., color cannot become a line, contrast cannot become a line, etc.), which means it has limited explanatory power.

We thank the reviewer for bringing up these excellent questions.

First, to test the alternative hypothesis of spatial attention, we fed a dot image into the image-computable model. We placed the dot where we suspect one might place their spatial attention, namely, at the edge of the stimulus that is tangent to the orientation of the grating. We generated the model response for three orientations and their combination by rotating and averaging. From Author response image 1 below, one can see that this model does not match the line-like representation we reported. Nonetheless, we would like to avoid making the argument that attention does not play a role. We strongly suspect that if one was attending to multiple places along a path that makes up a line, it would produce the results we observed. But there begins a circularity in the logic, where one cannot distinguish between attention to a line-like representation and a line of attention being the line-like representation.

**Author response image 1. sa3fig1:** Reconstruction maps for the dot image at the edge of 15°, 75°, 135°, and the combined across three orientation conditions.

Second, we remain agnostic to the question of whether fine-scale inhomogenous sampling of orientation selective neurons may drive some of the decoding results we report here. It is possible that our line-like representations are driven by neurons tuned to the sample orientation that have receptive fields that lie along the line. Here, we instead focus on testing the idea that WM decoding does not depend on aperture biases.

Finally, we agree with the reviewer that there is much more work to be done in this area. Our working hypothesis, that WM representations are abstractions of percepts, is admittedly based on Occam's razor and an appeal to efficient coding principles. We also agree that these results may not generalize to all forms of WM (eg, color). As always, there is a tradeoff between interpretability (visual spatial formats in retinotopically organized maps) and generalizability. Frankly, we have no idea how one might be able to test these ideas when subjects might be using the most common type of memory reformatting - linguistic representations, which are incredibly efficient.

Additional context:The working memory and perception tasks are rather different. In this case, the perception task does not require the subject to process the carrier orientation (which is largely occluded, and possibly not that obvious without paying attention to it), but attention is paid to contrast. In this scenario, stimulus energy may dominate the signal. In the WM task, subjects have to work out what orientation is shown to do the task. Given that the sensory stimulus in both tasks is brief (1.5s during memory encoding, and 2.5s total in the perceptual task), it would be interesting to look at decoding (and reconstructions) for the WM stimulus epoch. If abstraction (into a line) happens in working memory, then this perceptual part of the task should still be susceptible to aperture biases. It allows the authors to show that it is indeed during memory (and not merely the task or attentional state of the subject) that abstraction occurs.

Again, this is an excellent question. We used a separate perceptual task instead of the stimulus epoch as control mainly for two reasons. First, we used a control task in which participants had to process the contrast, not orientation, of the grating because we were concerned that participants would reformat the grating into a line-like representation to make the judgments. To avoid this, we used a task similar to the one used when previous researchers first found the stimulus vignetting effect (Roth et al., 2018). Again, our main goal was to try to focus on the bottom-up visual features. Second, because of the sluggishness of the BOLD response, combined with our task design (ie, memory delay always followed the target stimulus), we cannot disentangle the visual and memory responses that co-exist at this epoch. Any result could be misleading.

What's also interesting is what happens in the passive perceptual condition, and the fact that spatial reconstructions for areas beyond V1 and V2 (i.e., V3, V3AB, and IPS0-1) align with (implied) grating endpoints, even when an angular modulator is used (Figure 3C). Are these areas also "abstracting" the stimulus (in a line-like format)?

We agree these findings are interesting and replicate what we found in our previous paper (Kwak & Curtis, Neuron, 2022). We believe that these results do imply that these areas indeed store a reformatted line-like WM representation that is not biased by vignetting. We would like to extend a note of caution, however, because the decoding results in the higher order areas(V3AB, IPS0-1, etc) are somewhat poor (especially in comparison to V1, V2, V3) (see Figure 2).

**Reviewer #2:**
Summary:According to the sensory recruitment model, the contents of working memory (WM) are maintained by activity in the same sensory cortical regions responsible for processing perceptual inputs. A strong version of the sensory recruitment model predicts that stimulus-specific activity patterns measured in sensory brain areas during WM storage should be identical to those measured during perceptual processing. Previous research casts doubt on this hypothesis, but little is known about how stimulus-specific activity patterns during perception and memory differ. Through clever experimental design and rigorous analyses, Duan & Curtis convincingly demonstrate that stimulus-specific representations of remembered items are highly abstracted versions of representations measured during perceptual processing and that these abstracted representations are immune to aperture biases that contribute to fMRI feature decoding. The paper provides converging evidence that neural states responsible for representing information during perception and WM are fundamentally different, and provides a potential explanation for this difference.Strengths:(1) The generation of stimuli with matching vs. orthogonal orientations and aperture biases is clever and sets up a straightforward test regarding whether and how aperture biases contribute to orientation decoding during perception and WM. The demonstration that orientation decoding during perception is driven primarily by aperture bias while during WM it is driven primarily by orientation is compelling.(2) The paper suggests a reason why orientation decoding during WM might be immune to aperture biases: by weighting multivoxel patterns measured during WM storage by spatial population receptive field estimates from a different task the authors show that remembered but not actively viewed - orientations form "line-like" patterns in retinotopic cortical space.

We thank the reviewer for noting the strengths in our work.

Weaknesses:(1) The paper tests a strong version of the sensory recruitment model, where neural states representing information during WM are presumed to be identical to neural states representing the same information during perceptual processing. As the paper acknowledges, there is already ample reason to doubt this prediction (see, e.g., earlier work by Kok & de Lange, Curr Biol 2014; Bloem et al., Psych Sci, 2018; Rademaker et al., Nat Neurosci, 2019; among others). Still, the demonstration that orientation decoding during WM is immune to aperture biases known to drive orientation decoding during perception makes for a compelling demonstration.

We agree with the reviewer, and would add that the main problem with the sensory recruitment model of WM is that it remains underspecified. The work cited above and in our paper, and the results in this report is only the beginning of efforts to fully detail what it means to recruit sensory mechanisms for memory.

(2) Earlier work by the same group has reported line-like representations of orientations during memory storage but not during perception (e.g., Kwak & Curtis, Neuron, 2022). It's nice to see that result replicated during explicit perceptual and WM tasks in the current study, but I question whether the findings provide fundamental new insights into the neural bases of WM. That would require a model or explanation describing how stimulus-specific activation patterns measured during perception are transformed into the "line-like" patterns seen during WM, which the authors acknowledge is an important goal for future research.

We agree with the reviewer that perhaps some might see the current results as an incremental step given our previous paper. However, we would point out that researchers have been decoding memorized orientation from the early visual cortex for 15 years, and not one of those highly impactful studies had ever done what we did here, which was to test if decoded WM representations are the product of aperture biases. Not only do our results indicate that decoding memorized orientation is immune to these biases, but they critically suggest a reason why one can decode orientation during WM.

**Reviewer #3:**
Summary:In this work, Duan and Curtis addressed an important issue related to the nature of working memory representations. This work is motivated by findings illustrating that orientation decoding performance for perceptual representations can be biased by the stimulus aperture (modulator). Here, the authors examined whether the decoding performance for working memory representations is similarly influenced by these aperture biases. The results provide convincing evidence that working memory representations have a different representational structure, as the decoding performance was not influenced by the type of stimulus aperture.Strengths:The strength of this work lies in the direct comparison of decoding performance for perceptual representations with working memory representations. The authors take a well-motivated approach and illustrate that perceptual and working memory representations do not share a similar representational structure. The authors test a clear question, with a rigorous approach and provide convincing evidence. First, the presented oriented stimuli are carefully manipulated to create orthogonal biases introduced by the stimulus aperture (radial or angular modulator), regardless of the stimulus carrier orientation. Second, the authors implement advanced methods to decode the orientation information present, in visual and parietal cortical regions, when directly perceiving or holding an oriented stimulus in memory. The data illustrates that working memory decoding is not influenced by the type of aperture, while this is the case in perception. In sum, the main claims are important and shed light on the nature of working memory representations.

We thank the reviewer for noting the strengths in our work.

Weaknesses:I have a few minor concerns that, although they don't affect the main conclusion of the paper, should still be addressed.(1) Theoretical framing in the introduction: Recent work has shown that decoding of orientation during perception does reflect orientation selectivity, and it is not only driven by the stimulus aperture (Roth, Kay & Merriam, 2022).

Excellent point, and similar to the point made by Reviewer 1. We now adjust our text and cite the paper in the Introduction.

Below, we paste our response to Reviewer 1:

“Second, we remain agnostic to the question of whether fine-scale inhomogenous sampling of orientation selective neurons may drive some of the decoding we report here. It is possible that our line-like representations are driven by neurons tuned to the sample orientation that have receptive fields that lie along the line. Here, we instead focus on testing the idea that WM decoding does not depend on aperture biases.”

(2) Figure 1C illustrates the principle of how the radial and angular modulators bias the contrast energy extracted by the V1 model, which in turn would influence orientation decoding. It would be informative if the carrier orientations used in the experiment were shown in this figure, or at a minimum it would be mentioned in the legend that the experiment used 3 carrier orientations (15{degree sign}, 75{degree sign}, 135{degree sign}) clockwise from vertical. Related, when trying to find more information regarding the carrier orientation, the 'Stimuli' section of the Methods incorrectly mentions that 180 orientations are used as the carrier orientation.

We apologize for not clearly indicating the stimulus features in the figure. Now, we added the information about the target orientations in Figure 1C legend. Also, we now corrected in the Methods section the mistakes about the carrier orientation and the details of the task. Briefly, participants were asked to use a continuous report over 180 orientations. We now clarify that “We generated 180 orientations for the carrier grating to cover the whole orientation space during the continuous report task.”

(3) The description of the image computable V1 model in the Methods is incomplete, and at times inaccurate. (i) The model implements 6 orientation channels, which is inaccurately referred to as a bandwidth of 60{degree sign} (should be 180/6=30). (ii) The steerable pyramid combines information across phase pairs to obtain a measure of contrast energy for a given stimulus. Here, it is only mentioned that the model contains different orientation and spatial scale channels. I assume there were also 2 phase pairs, and they were combined in some manner (squared and summed to create contrast energy). Currently, it is unclear what the model output represents. (iii) The spatial scale channel with the maximal response differences between the 2 modulators was chosen as the final model output. What spatial frequency does this channel refer to, and how does this spatial frequency relate to the stimulus?

(i) First, we thank the reviewer for pointing out this mistake since the range of orientations should be 180deg instead of 360deg. We corrected this in the revised version.

(ii) Second, we apologize for not being clear. In the second paragraph of the “Simulate model outputs” section, we wrote,

“For both types of stimuli, we used three target orientations (15°, 75°, and 135° clockwise from vertical), which had two kinds of phases for both the carriers and the modulators. We first generated the model’s responses to each target image separately, then averaged the model responses across all phases for each orientation condition.”

We have corrected this text by now writing,

from vertical), two phases for the carrier (0 or π), and two phases for the modulator (sine “For both types of stimuli, we used three target orientations (15°, 75°, and 135° clockwise from vertical), two phases for the carrier (0 or π), and two phases for the modulator (sine or cosine phase). We first generated the model responses to each phase condition separately, then averaged them across all phases for each orientation condition.”

(iii) Third and again we apologize for the misunderstanding. Since both modulated gratings have the same spatial frequency, the channel with the largest response should be equal to the spatial frequency of the stimulus. We corrected this by now writing,

“For the final predicted responses, we chose the subband with maximal responses (the 9th level), which corresponds to the spatial frequency of the stimulus (Roth, Heeger, and Merriam 2018).”

(4) It is not clear from the Methods how the difficulty in the perceptual control task was controlled. How were the levels of task difficulty created?

Apologies for not being clear. The task difficulty was created by setting the contrast differences between the two stimuli. The easiest level is choosing the first and the last contrast as pairs, while the hardest level is choosing the continuous two contrasts. We added these sentences

“The contrast for each stimulus was generated from a predefined set of 20 contrasts uniformly distributed between 0.5 and 1.0 (0.025 step size). We created 19 levels of task difficulty based on the contrast distance between the two stimuli. Thus, the difficulty ranged from choosing contrast pairs with the largest difference (0.5, easiest) to contrast pairs with the smallest difference (0.025, hardest). Task difficulty level changed based on an adaptive, 1-up-2-down staircase procedure (Levitt 1971) to maintain performance at approximately 70% correct.”

**Recommendations For The Authors**

**(Reviewer #1):**
(1) If the black circle (Fig 3A & C) is the stimulus size, and the stimulus (12º) is roughly half the size of the entire screen (24.8º), then how are spatial reconstructions generated for parts of the visual field that fall outside of the screen? I am asking because in Figure 3 the area over which spatial reconstructions are plotted has a diameter at least 3 times the diameter of that black circle (the stimulus). I'm guessing this is maybe possible when using a very liberal fitting approach to prf's, where the center of a prf can be outside of the screen (so you'd fit a circle to an elongated blob, assuming that blob is the edge of a circle, or something). Can you really reliably estimate that far out into visual space/ extrapolate prf's that exist in a part of the space you did not fully map (because it's outside of the screen)?

We thank the reviewer for pointing out this confusing issue.

First, the spatial construction map has a diameter 3 times the diameter of the stimulus because we included voxels whose pRF eccentricities were within 20º in the reconstruction, the same as Kwak & Curtis, 2022. There are reasons for doing so. First, while the height of the screen is 24.8º, the width of the screen is 44º. Thus, it is possible to have voxels whose pRF eccentricities are >20º. Second, for areas outside the height boundaries, there might not be pRF centers, but the whole pRF Gaussian distributions might still cover the area. Moreover, when creating the final map combined across three orientation conditions, we rotated them to be centered vertically, which then required a 20x20º square. Finally, inspecting the reconstruction maps, we noticed that the area that was twice the stimulus size (black circle) made very little contributions to the reconstructions. Therefore, the results depicted in Figure 3A&C are justified, but see the next comment and our response.

(2) Is the quantification in 3B/C justified? The filter line uses a huge part of visual space outside of the stimulus (and even the screen). For the angular modulator in the "perception" condition, this means that there is no peak at -90/90 degree. But if you were to only use a line that is about the size of the stimulus (a reasonable assumption), it would have a peak at -90/90 degree.

This is an excellent question. We completely agree that it is more reasonable to use filter lines that have the same size (12º) as the stimulus instead of the whole map size (40º). Based on the feedback from the Reviewer, we redid the spatial reconstruction analyses and now include the following changes to Figure 3.

(1) We fitted the lines using pixels only within the stimulus. In Figure 3A and Figure 3C, we now replaced the reconstruction maps.

(2) We added the color bar in Figure 3A.

(3) We regenerated the filtered responses and calculated the fidelity results by using line filters with the stimulus size. We replaced the filtered responses and fidelity results in Figure 3B and Figure 3D. With the new analysis, as anticipated by the Reviewer, we now found peaks at -90/90 degrees for the angular modulated gratings in the perceptual control task in V1 and V2. Thank you Reviewer 1!!!!

(4) We also made corresponding changes in the Supplementary Figure S4 and S5, as well as the statistical results in Table S4 and S5.

(5) In the “Methods” section, we added “within the stimulus size” for both “fMRI data analysis: Spatial reconstruction” and “Quantification and statistical analysis” subsections.

(3) Figure 4 is nice, but not exactly quantitative. It does not address that the reconstructions from the perceptual task are hugging the stimulus edges much more closely compared to the modeled map. Conversely, the yellow parts of the reconstructions from the delay fan out much further than those of the model. The model also does not seem to dissociate radial/angular stimuli, while in the perceptual data the magnitude of perceptual reconstruction is clearly much weaker for angular compared to radial modulator.

We thank the reviewer for this question. First, we admit that Figure 4 is more qualitative than quantitative. However, we see no alternative that better depicts the similarity in the model prediction and the fMRI results for the perceptual control and WM tasks. The figure clearly shows the orthogonal aperture bias. Second, we agree that aspects of the observed fMRI results are not perfectly captured by the model. This could be caused by many reasons, including fMRI noise, individual differences, etc. Importantly, different modulators induce orthogonal aperture bias in the perceptual but not the WM task, and therefore does not have a major impact on the conclusions.

(4) The working memory and perception tasks are rather different. In this case, the perception task does not require the subject to process the carrier orientation (which is largely occluded, and possibly not that obvious without paying attention to it), but attention is paid to contrast. In this scenario, stimulus energy may dominate the signal. In the WM task, subjects have to work out what orientation is shown to do the task. Given that the sensory stimulus in both tasks is brief (1.5s during memory encoding, and 2.5s total in the perceptual task), it would be interesting to look at decoding (and reconstructions) for the WM stimulus epoch. If abstraction (into a line) happens in working memory, then this perceptual part of the task should still be susceptible to aperture biases. It allows the authors to show that it is indeed during memory (and not merely the task or attentional state of the subject) that abstraction occurs.

We addressed the same point in the response for Reviewer 1, “additional context” section.

Recommendations for improving the writing:(1) The main text had too little information about the Methods. Of course, some things need not be there, but others are crucial to understanding the basics of what is being shown. For example, the main text does not describe how many orientations are used (well... actually the caption to Figure 1 says there are 2: horizontal and vertical, which is confusing), and I had to deduce from the chance level (1/3) that there must have been 3 orientations. Also, given how important the orthogonality of the carrier and modulator are, it would be good to have this explicit (I would even want an analysis showing that indeed the two are independent). A final example is the use of beta weights, and for delay period decoding only the last 6s (of the 12s delay) are modeled and used for decoding.

We thank the reviewer for identifying aspects of the manuscript that were confusing. We made several changes to the paper to clarify these details.

First, we added the information about the orientations we used in the caption for Figure 1 and made it clear that Figure 1C is just an illustration using vertical/horizontal orientations. Second, the carrier and the modulator are different in many ways. For example, the carrier is a grating with orientation and contrast information, while the modulator is the aperture that bounds the grating without these features. Their phases are orthogonal, and we added this in the second paragraph of the “Stimuli” section. Last, in the main text and the captions, we now denote “late delay” when writing about our procedures.

(2) Right under Figure 3, the text reads "angular modulated gratings produced line-like representations that were orthogonal carrier orientation reflecting the influence of stimulus vignetting", but the quantification (Figure 3D) does not support this (there is no orthogonal "bump" in the filtered responses from V1-V3, and one aligned with the carrier orientation in higher areas).

This point was addressed in the “recommendations for the authors (Reviewer 1), point 2” above.

Minor corrections to text and figures:(1) Abstract: "are WM codes" should probably be "WM codes are".

We prefer to keep “are WM codes” as it is grammatically correct.

(2) Introduction: Second sentence 2nd paragraph: representations can be used to decode representations? Or rather voxel patterns can be used...

Changed to “On the one hand, WM representations can be decoded from the activity patterns as early as primary visual cortex (V1)...”

(3) Same paragraph: might be good to add more references to support the correlation between V1 decoding and behavior. There's an Ester paper, and Iamchinina et al. 2021. These are not trial-wise, but trial-wise can also be driven by fluctuating arousal effects, so across-subject correlations help fortify this point.

We added these two papers as references.

(4) Last paragraph: "are WM codes" should probably be "WM codes are".

See (1) above.

(5) Figure 1B & 2A caption: "stimulus presenting epoch" should probably be "stimulus presentation epoch".

Changed to “stimulus epoch”.

(6) Figure 1C: So this is very unclear, to say stimuli are created using vertical and horizontal gratings (when none of the stimuli used in the experiment are either).

We solved and answered this point in response to Reviewer 3, point 2.

(7) Figure 2B caption "cross" should probably be "across".

We believe “cross” is fine since cross here means cross-decoding.

(8) Figure 3A and C are missing a color bar, so it's unclear how these images are generated (are they scaled, or not) and what the BOLD values are in each pixel.

All values in the map were scaled to be within -1 to 1. We added the color bar in both Figure 3 and Figure 4.

(9) Figure 3B and D (bottom row) are missing individual subject data.

We use SEM to indicate the variance across subjects.

(10) Figure D caption: "early (V1 and V2)" should probably be "early areas (V1 and V2)".

Corrected.

(11) Methods, stimuli says "We generated 180 orientations for the carrier grating to cover the whole orientation space." But it looks like only 3 orientations were generated, so this is confusing.

We solved and answered this point in response to Reviewer 3, point 2.

(12) Further down (fMRI task) "random jitters" is probably "random jitter"

Corrected.